# RAPID[3]: Tri-Level Reinforced Acceleration Policies for Diffusion Transformer

**Wangbo Zhao**[1*]    **Yizeng Han**[2]    **Zhiwei Tang**[2,3,4]    **Jiasheng Tang**[2,3†]    **Pengfei Zhou**[1]
**Kai Wang**[1]    **Bohan Zhuang**[2,4]    **Zhangyang Wang**[5]    **Fan Wang**[2]    **Yang You**[1†]

[1]National University of Singapore    [2]DAMO Academy, Alibaba Group    [3]Hupan Lab
[4]ZIP Lab, Zhejiang University    [5]The University of Texas at Austin

## ABSTRACT

Diffusion Transformers (DiTs) excel at visual generation yet remain hampered by slow sampling. Existing training-free accelerators—step reduction, feature caching, and sparse attention—enhance inference speed but typically rely on a uniform heuristic or manually designed adaptive strategy for all images, leaving quality on the table. Alternatively, dynamic neural networks offer per-image adaptive acceleration, but their high fine-tuning costs limit broader applicability. To address these limitations, we introduce **RAPID³**: *Tri-Level Reinforced Acceleration PolIcies for Diffusion Transformer* framework that delivers image-wise acceleration with *zero* updates to the base generator. Specifically, three lightweight policy heads—*Step-Skip*, *Cache-Reuse*, and *Sparse-Attention*—observe the current denoising state and independently decide their corresponding speed-up at each timestep. All policy parameters are trained online via Group Relative Policy Optimization (GRPO) while the generator remains frozen. Meanwhile, an adversarially learned discriminator augments the reward signal, discouraging reward hacking by boosting returns only when generated samples stay close to the original model's distribution. Across state-of-the-art DiT backbones including Stable Diffusion 3 and FLUX, RAPID³ achieves nearly **3×** faster sampling with competitive generation quality. The code is publicly available at https://github.com/NUS-HPC-AI-Lab/RAPID3.

## 1 INTRODUCTION

Diffusion Transformers (DiTs) have emerged as the dominant backbone for high-fidelity visual generation thanks to their scalability and strong generalization (Peebles & Xie, 2023; Bao et al., 2023). They now underpin state-of-the-art systems in diverse downstream tasks—image synthesis (Chen et al., 2023; Esser et al., 2024; Labs, 2024), video generation (OpenAI, 2024; team, 2025), and controllable editing (Xiao et al., 2024; Feng et al., 2025). Despite this progress, DiTs remain inefficient during inference, requiring multiple denoising steps with computationally intensive blocks on large latent maps, making real-world deployment challenging.

A first family of training-free accelerators (Figure 1 (a)) tackles this cost by hard-coded heuristics: reducing the step count (Song et al., 2020; Lu et al., 2022a;b; Park et al., 2024), reusing intermediate features (Ma et al., 2024b; Chen et al., 2024; Selvaraju et al., 2024; Liu et al., 2024), or sparsifying attention (Zhang et al., 2025a; Yuan et al., 2024b). These techniques preserve the frozen generator but apply a uniform or manually designed adaptive policy to every image and timestep; quality therefore fluctuates, and conservative settings are adopted to avoid artifacts, sacrificing potential speed-ups. A second family—dynamic neural networks trained with fine-tuning (Figure 1 (b))—learns routers that adapt width, depth, or spatial resolution on the fly (Han et al., 2021; Ganjdanesh et al., 2024; Zhao et al., 2024; You et al., 2024; Anagnostidis et al., 2025; Zhao et al., 2026). While these methods offer adaptability and high performance, they require extensive optimization on large-scale image–text datasets, making them impractical for many large generators or proprietary models.

---

*Work done during Wangbo's internship at DAMO Academy, Alibaba Group. wangbo.zhao96@gmail.com
†Corresponding author.

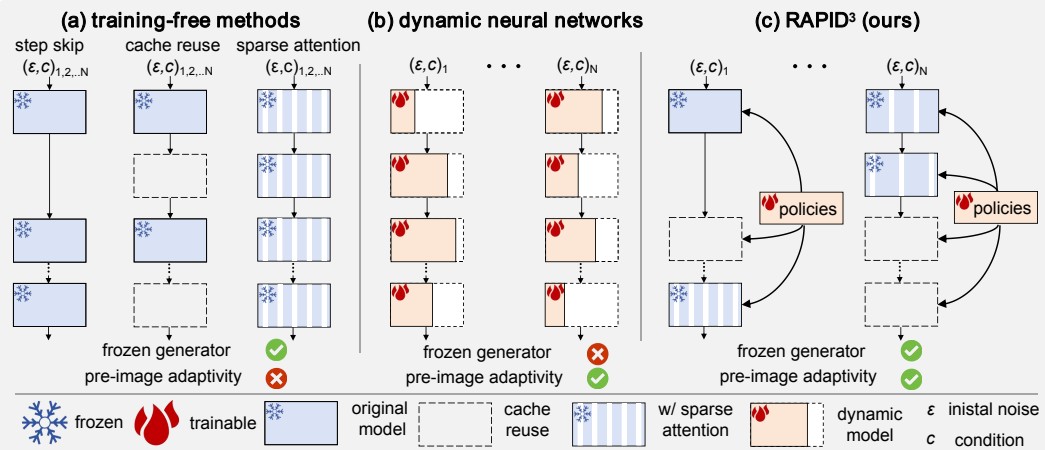

Figure 1: Accelerating Diffusion Transformers: **(a) Training-free methods** primarily use uniform or manually designed rules (*e.g.*, step skip, cache reuse, or sparse attention) for all images and timesteps, offering speed but little adaptivity. **(b) Dynamic fine-tuned models** learn routers that tailor acceleration to each input but require costly fine-tuning of the generator's parameters. **(c) Proposed RAPID[3]** keeps the generator frozen and equips it with three lightweight policy heads—*Step-Skip*, *Cache-Reuse*, and *Sparse-Attention*—trained via reinforcement learning to make action decisions based on the latent representations, timestep, and text prompt information.

Sampling in a diffusion transformer, however, is naturally a Markov decision process: its forward pass unfolds as a sequence of denoising steps. If we consider the latent generated at each step—along with its timestep index and prompt embedding—as the *state*, then the selection of the next timestep can be treated as a *continuous action*, while decisions on whether to reuse cached features or apply sparse attention are regarded as *discrete actions*. Since the generator's weights remain fixed, the outcome of any action is deterministic and fully predictable by the model itself, eliminating the need to learn environment dynamics. Episodes are inherently brief, typically spanning multiple sampling steps, and conclude with a complete image whose quality can be quantitatively assessed post hoc, enabling a straightforward reward signal that balances image fidelity against computational cost.

Our **key insight** is that the DiT inference setting, when viewed through the above lens, sits squarely in the comfort zone of modern policy-optimization methods: a concise but expressive action space; a stationary and deterministic environment; an inexpensive simulator (the frozen DiT) that delivers abundant rollouts; and a scalar reward that directly reflects the user's objective. By leveraging this structure, we can train lightweight policies to dynamically determine which acceleration strategy to invoke—enabling adaptivity without modifying the generator's parameters. The recent TPDM (Ye et al., 2024) relates to our idea: it achieves data-dependent step skipping via reinforcement learning (RL), while neglecting the redundant computation from the intra-timestep perspective.

Motivated by this observation, we present **RAPID[3]** (Figure 1 (c))—*Tri-Level Reinforced Acceleration PolIcies for Diffusion Transformer*. It attaches three policy heads—*Step-Skip*, *Cache-Reuse*, and *Sparse-Attention*—to a frozen DiT. Each head observes summaries of the latent, timestep, or prompt, and independently decides corresponding acceleration strategies. All policy parameters are trained online with Group Relative Policy Optimisation (GRPO) (Shao et al., 2024). Moreover, instead of using existing evaluation metrics to build the reward function, we train a discriminator adversarially to enhance the reward for samples that remain close to the original generator's distribution, which prevents the reward hacking problem (Skalse et al., 2022).

Experimental results show that RAPID[3] achieves nearly $3\times$ acceleration for Stable Diffusion 3 (Esser et al., 2024) and FLUX (Labs, 2024), while maintaining competitive visual quality. By updating only the policy head, which constitutes merely 0.025% of the generator's parameters, its training process relies solely on readily available text prompts and and consumes only 1% of the GPU hours required for training dynamic neural networks (Zhao et al., 2026).

## 2 RELATED WORK

**Diffusion transformers.** Diffusion models (Sohl-Dickstein et al., 2015; Ho et al., 2020; Albergo et al., 2023) have achieved remarkable success in visual generation tasks (Rombach et al., 2022; OpenAI, 2024). Early diffusion models (Ho et al., 2020; Dhariwal & Nichol, 2021; Rombach et al., 2022) are primarily built on U-Net (Ronneberger et al., 2015). However, Transformer-based architectures (Vaswani et al., 2017) have gradually replaced U-Net as the foundation for these models. DiT (Peebles & Xie, 2023) represents one of the first attempts to integrate Transformers into the diffusion process. Concurrently, U-ViT (Bao et al., 2023) combines the advantages of U-Net's skip connections with the capabilities of Transformer architectures. Building on DiT, PixArt-$\alpha$ expands its capabilities to text-to-image generation, while SD3 (Esser et al., 2024) and FLUX (Labs, 2024) further demonstrate its scalability. Beyond image generation, Transformer-based diffusion models, such as Sora (OpenAI, 2024) and WanX (WanTeam et al., 2025), have also shown significant promise in video generation. These progresses in foundation models have also facilitated the development of various applications, such as image and video editing (Ma et al., 2024c;d; 2025e;d;a). Despite these achievements, diffusion models continue to suffer from high computational demands and time-consuming generation. The proposed framework addresses this limitation by adaptively adjusting computation for each image generation process, enabling more efficient visual generation.

**Diffusion transformer acceleration.** To enhance inference efficiency of DiTs, researchers have explored training-free techniques such as reducing steps (Song et al., 2020; Lu et al., 2022a; Park et al., 2024; Lu et al., 2022b), feature caching (Ma et al., 2024b; Chen et al., 2024; Liu et al., 2024; Huang et al., 2024), and sparse attention mechanisms (Zhang et al., 2025a;b; Xi et al., 2025). However, these approaches typically use uniform strategies for all images and the entire generation process or depend on manually designed heuristics, limiting their generalization. Inspired by the efficiency gains of dynamic neural networks (Han et al., 2021; Verma et al., 2024; Zhao et al., 2024; 2025a; Wang et al., 2025), some studies (Ganjdanesh et al., 2024; Zhao et al., 2025b; 2026) introduce dynamic architectures to improve efficiency, while others (You et al., 2024; Anagnostidis et al., 2025) dynamically reduce computation along the spatial dimension. However, these methods require fine-tuning diffusion models, which imposes a significant training burden and becomes impractical when training data is limited. Few-step distillation techniques (Luo et al., 2023; Yan et al., 2024; Lin et al., 2024; Yin et al., 2024; Shao et al., 2025) accelerate inference but often require extensive parameter tuning and training of around 100M additional parameters, even with methods like LoRA (Hu et al., 2022). In contrast, our approach trains lightweight policy heads with only 3M additional parameters, adaptively selecting acceleration strategies while keeping DiT parameters frozen. This ensures both parameter and data efficiency, offering a more practical and elegant solution.

**Reinforcement learning in diffusion models.** The success of reinforcement learning (RL) in large language models (LLMs) (Ouyang et al., 2022; Shao et al., 2024; Guo et al., 2025; Team, 2025) has encouraged researchers to explore it in diffusion models. DDPO (Black et al., 2023) views denoising as a multi-step decision-making task, aligning generated images with downstream objectives. Similarly, DPOK (Fan et al., 2023) demonstrates the effectiveness of RL in fine-tuning diffusion models, achieving superior text-image alignment and improved image fidelity. Some approaches (Wallace et al., 2024; Yuan et al., 2024a) have also explored the alignment of generation quality with human preferences in an offline RL framework. However, these methods primarily focus on enhancing the generation quality rather than accelerating the generation speed. TPDM (Ye et al., 2024) alleviates this problem by learning an efficient noise scheduler, without exploring its incorporation with other acceleration techniques. Our approach leverages RL to learn policy heads that dynamically determine step-skipping, cache-reuse, and sparse-attention strategies at each timestep, adaptively improving the generation efficiency.

## 3 METHODOLOGY

We begin with foundational concepts of DiT and RL in Section 3.1. Then, we present the proposed tri-level reinforced acceleration policies and their corresponding policy heads in Section 3.2 and Section 3.3, respectively. Section 3.4 details the adversarial reinforcement learning within the the proposed framework. A comprehensive overview of the methodology is provided in Figure 2.

## 3.1 PRELIMINARY

**Architecture of diffusion transformers.** Diffusion Transformers (DiTs) (Peebles & Xie, 2023; Bao et al., 2023; Chen et al., 2023; Esser et al., 2024; Labs, 2024) has demonstrated remarkable advancements in visual generation thanks to its scalable architecture. It generally consists of a stack of layers, each of which integrates a self-attention (SA) block and a multi-layer perceptron (MLP) block. The operation of a DiT block can be roughly expressed as:

$$\mathbf{x}_t^{l+1}, \mathbf{c}_t^{l+1} = \mathcal{F}_{\text{LAYER}}^l(\mathbf{x}_t^l, \mathbf{c}_t^l) = \mathcal{F}_{\text{MLP}}^l(\mathcal{F}_{\text{SA}}^l(\mathbf{x}_t^l, \mathbf{c}_t^l)), \tag{1}$$

where $\mathbf{x}_t^l \in \mathbb{R}^{N \times C}$ denotes the input to the $l$-th layer at timestep $t$. Here, $N$ represents the number of tokens and $C$ the channel dimension. The condition embedding $\mathbf{c}_t^l$ combines timestep information with text-based guidance, which are critical for generation.

**Reinforcement learning with Group Relative Policy Optimization.** To improve the performance and alignment of LLMs with human preferences, RL techniques such as Proximal Policy Optimization (PPO) (Schulman et al., 2017) are widely applied during fine-tuning. To reduce the training cost, Group Relative Policy Optimization (GRPO) (Shao et al., 2024) is proposed, which leverages a group-based strategy to estimate advantages. Specifically, for a given question $q$ from the training set, GRPO samples a group of output answers $\{o_i\}_{i=1}^G$ from a LLM model $\pi_{\theta_{\text{old}}}$ with fixed parameters. The target policy $\pi_\theta$ is then updated by maximizing the following objective: $\mathcal{J}_{\text{GRPO}}(\theta) = \frac{1}{G} \sum_{i=1}^G \left( \min \left( \frac{\pi_\theta(o_i|q)}{\pi_{\theta_{\text{old}}}(o_i|q)} A_i, \text{clip} \left( \frac{\pi_\theta(o_i|q)}{\pi_{\theta_{\text{old}}}(o_i|q)}, 1-\varepsilon, 1+\varepsilon \right) A_i \right) - \beta \mathbb{D}_{\text{KL}} \right)$, where $\varepsilon$ and $\beta$ are hyperparameters. The term $\mathbb{D}_{\text{KL}}$ represents a KL-divergence penalty between the current model $\pi_\theta$ and the pre-trained reference model $\pi_{\text{ref}}$. The advantage $A_i$ is computed from a group of rewards $\{r_i\}_{i=1}^G$ as $A_i = \frac{r_i - \text{mean}(\{r_1, r_2, \cdots, r_G\})}{\text{std}(\{r_1, r_2, \cdots, r_G\})}$. Due to its simplicity and effectiveness, we integrate GRPO into the generation process of diffusion transformers to train policy heads, as discussed later in Section 3.4.

## 3.2 TRI-LEVEL REINFORCED ACCELERATION POLICIES

As previously outlined, our goal is to find policies that dynamically selects acceleration strategies for visual generation. To achieve this, we define three levels of candidate strategies to accelerate from model-external to model-internal perspectives: **Step-Skip**, **Cache-Reuse**, and **Sparse-Attention**. These strategies are effective and training-free acceleration techniques, but existing approaches often apply them with uniform or manually designed adaptive policies.

**Level-1: Step-Skip.** The diffusion Transformer's generation involves a multi-timestep schedule, where the timestep decreases from $t_T$ to $t_0$. This process' efficiency is critically dependent on the required of timestep number. A natural idea is to reduce the needed timesteps. However, prior methods (Song et al., 2020; Lu et al., 2022a; Park et al., 2024; Lu et al., 2022b) employ fixed schedules, *ignoring that different images may require varying timesteps*. For instance, highly detailed images often demand more timesteps to achieve high-quality outputs, whereas simpler images can be generated effectively with fewer steps. Applying a uniform schedule across all cases risks degrading both *generation quality* and *computational efficiency*.

To achieve inference-time dynamic timestep jumping, we define a policy $\mathcal{P}^{\text{step}}$ that selects the current timestep $t$ based on the state of the previous timestep $t_{\text{prev}}$. Specifically, let $\mathcal{G}(\cdot)$ denote the forward computation of DiT, $\text{sampler}(\cdot)$ represent the diffusion sampling function, and $\mathbf{X}_{t_{\text{prev}}}$ indicate the latent feature at the previous timestep. The feature at the current timestep, $\mathbf{X}_t$, can be formulated as

$$\mathbf{X}_t \leftarrow \text{sampler}(\mathbf{X}_{t_{\text{prev}}}, \mathcal{G}(\mathbf{X}_{t_{\text{prev}}}, t_{\text{prev}}), t). \tag{2}$$

This approach allows the generation process to adjust the number of required timesteps per input, enabling *varying computational resource allocation* depending on the the target complexity. As a result, the diffusion process achieves a balance between generation quality and efficiency.

**Level-2: Cache-Reuse.** Feature caching mechanisms (Ma et al., 2024b; Chen et al., 2024; Selvaraju et al., 2024; Liu et al., 2024) leverage temporal coherence in diffusion processes by reusing computed feature maps across consecutive timesteps. A prevalent approach in diffusion transformers in-

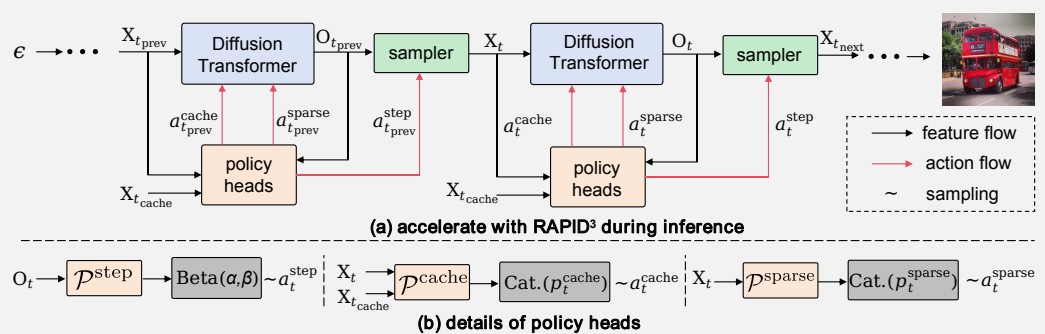

Figure 2: **Overview of RAPID³.** (a) Accelerate generation with RAPID³ during inference. (b) The details of policy heads $\{\mathcal{P}^{\text{step}}, \mathcal{P}^{\text{cache}}, \mathcal{P}^{\text{sparse}}\}$.

volves caching the residual components of feature maps (Chen et al., 2024; Liu et al., 2024) - specifically, the difference between a model's input and output. For timestep $t$, this enables computational simplification through reuse of the residual cached at timestep $t_{\text{cache}}$, expressed as $\mathbf{O}_t \approx \mathbf{X}_t + \Delta_{t_{\text{cache}}}$, where $\mathbf{O}_t$ denotes the output of the diffusion transformer and $\Delta_{t_{\text{cache}}} = \mathcal{G}(\mathbf{X}_{t_{\text{cache}}}, t_{\text{cache}}) - \mathbf{X}_{t_{\text{cache}}}$ represents the cached residual.

Two fundamental challenges emerge in practical implementations: a) The trade-off between acceleration and generation quality exhibits strong dependence on the temporal interval between cache updates. Current approaches relying on fixed or heuristic update intervals demonstrate limited generalizability, particularly when combined with other acceleration techniques. b) Generating different images may require distinct caching schedules, posing challenges to designing and optimization.

Hence, the 2nd level of our objective centers on developing an adaptive policy $\mathcal{P}^{\text{cache}}$ that *dynamically selects optimal caching strategies* per timestep $t$. Formally, the policy determines the computation path through the following conditional operation:

$$\mathbf{O}_t = \mathcal{G}(\mathbf{X}_t, t) \text{ if update cache}, \quad \mathbf{X}_t + \Delta_{t_{\text{cache}}} \text{ if reuse cache}. \tag{3}$$

If the policy model determines to update the cache, we will conduct the computation $\mathcal{G}(\mathbf{X}_t, t)$ as the original diffusion transformer and update the cache to $\Delta_t$.

**Level-3: Sparse-Attention.** The quadratic computational complexity of self-attention mechanisms in transformers ($O(n^2)$ for $n$ tokens) poses significant latency challenges for high-resolution image generation. To mitigate this bottleneck, recent work (Zhang et al., 2025a; Yuan et al., 2024b) has integrated sparse attention mechanisms into diffusion transformers through conditional computation of attention weights. These approaches prune the computation in attention maps by employing a hyperparameter $\theta \in^{L \times H}$ to control sparsity for each layer and attention head, where $L$ and $H$ represent the number of layers and attention heads, respectively. Existing implementations, such as SpargeAttn (Zhang et al., 2025a), search for $\theta$ over the entire diffusion process of DiT and keep it fixed across all timesteps during generation.

However, this static configuration ignores the evolution of attention patterns during the diffusion process. During the diffusion process, different timesteps exhibit distinct properties and may vary in their sensitivity to sparsity. Fixing the $\theta$ could sacrifice potential speed-up or lead to quality degradation. More importantly, the $\theta$ optimized for the original DiT may become unsuitable when DiT is combined with other acceleration methods.

To address this, we propose a dynamic sparsity policy $\mathcal{P}^{\text{sparse}}$, to *adaptively identify a optimal hyperparemter* for the current timestep, $\theta_t \in^{L \times H}$, from a series of pre-defined candidate hyperparemters $\{\theta_1, \theta_2, ...\theta_{N_{\text{sparse}}}\}$. The original self-attention block can then be replaced with sparse attention using the identified hyperparameter, which is formulated as:

$$\mathcal{F}_{\text{SA}}^l(\mathbf{x}_t^l, \mathbf{c}_t^l) \xrightarrow{\mathcal{P}^{\text{sparse}}} \mathcal{F}_{\text{SA}_{\text{sparse}}}^l(\mathbf{x}_t^l, \mathbf{c}_t^l; \theta_t^l). \tag{4}$$

This approach enables the adaptation of computation sparsity on a per-timestep basis.

### 3.3 DESIGN OF POLICY HEADS

The learning of our tri-level acceleration framework involves joint optimization of three policy heads $\mathcal{P}^{\text{step}}, \mathcal{P}^{\text{cache}}, \mathcal{P}^{\text{sparse}}$. They determine the acceleration strategy based on the state at the current timestep $t$. Each policy head begins with a convolution layer for feature projection, followed by AdaLN (Perez et al., 2018) integrating the condition embedding $c_t$. The output is finally pooled and fed to the corresponding linear head for action prediction. Each policy head is detailed as follows:

**a) Step-Skip policy** $\mathcal{P}^{\text{step}}$. It predicts jump steps via parametric distribution learning:

$$\mathcal{P}^{\text{step}} : \mathbf{O}_t \mapsto [\alpha, \beta], \quad a_t^{\text{step}} \sim \text{Beta}(\alpha, \beta), \quad t_{\text{next}} = \lfloor t \cdot a_t^{\text{step}} \rfloor, \tag{5}$$

where $\mathbf{O}_t$ denotes the output of the diffusion transformer at the current timestep $t$. Inspired by (Ye et al., 2024), $\mathcal{P}_{\text{step}}$ uses a linear layer to regress two values, $\alpha$ and $\beta$, which parameterize a Beta distribution. A value from this distribution is subsequently used to compute the next timestep $t_{\text{next}}$.

**b) Cache-Reuse policy** $\mathcal{P}^{\text{cache}}$. This discrete decision network determines whether to perform computation or reuse the cached result based on the discrepancy between the inputs at the current timestep $\mathbf{X}_t$ and the last cached timestep $\mathbf{X}_{t_{\text{cache}}}$. This process can be expressed as:

$$\mathcal{P}^{\text{cache}} : \mathbf{X}_t - \mathbf{X}_{t_{\text{cache}}} \mapsto \mathbf{p}_t^{\text{cache}} \in \mathbb{R}^2, \quad a_t^{\text{cache}} \sim \text{Categorical}(\mathbf{p}_t^{\text{cache}}) \tag{6}$$

with action semantics: $a_t^{\text{cache}} = 0$ indicates that computation should be performed at the current timestep to update the cache, while $a_t^{\text{cache}} = 1$ correspond to reusing the cache.

**c) Sparse-Attention policy** $\mathcal{P}^{\text{sparse}}$. This policy adapts computation through discrete choices:

$$\mathcal{P}^{\text{sparse}} : \mathbf{X}_t \mapsto \mathbf{p}_t^{\text{sparse}} \in \mathbb{R}^{1+N_{\text{sparse}}}, \quad a_t^{\text{sparse}} \sim \text{Categorical}(\mathbf{p}_t^{\text{sparse}}), \tag{7}$$

where $a_t^{\text{sparse}} = 0$ indicates full attention where no sparsity is applied, while $a_t^{\text{sparse}} \in [1, N_{\text{sparse}}]$ denotes applying sparse attention with different $\theta$ values that progressively increase the sparsity ratios. Here, $N_{\text{sparse}}$ is set to 3 by default. Although we experiment with additional candidate hyperparameters, we observe that an overly aggressive sparsity strategy significantly degrades the generation quality. Notably, $\mathcal{P}_t^{\text{sparse}}$ remains inactive when $\mathcal{P}_t^{\text{cache}}$ decides to reuse the cache.

### 3.4 ADVERSARIAL REINFORCEMENT LEARNING

To train the policy heads, we propose an adversarial reinforcement learning framework, inspired by adversarial training (Goodfellow et al., 2020; Lin et al., 2024; Sauer et al., 2024; Lin et al., 2025). This approach iteratively trains the policy heads alongside a discriminator model, effectively addressing the challenging issue of reward hacking. The training pipeline is outlined in Algorithm 1.

**Training policy heads with reinforcement learning.** To train the policy heads, we adopt the diffusion transformer $\mathcal{G}$ equipped with policy heads $\{\mathcal{P}^{\text{step}}, \mathcal{P}^{\text{cache}}, \mathcal{P}^{\text{sparse}}\}$ to sample a group of images $\{I_i\}_{i=1}^G$, based on the same text prompt. Subsequently, the pre-trained image reward model $\mathcal{Q}$ is employed to evaluate the quality and alignment of each image with the given text condition $c$, producing scores $\{q_i\}_{i=1}^G$. Simultaneously, the discriminator model $\mathcal{D}$ (details provided later), is used to estimate the likelihood that a generated image originates from the diffusion transformer without acceleration, producing scores $\{d_i\}_{i=1}^G$. This framework encourages the policy heads to identify acceleration strategies that ensure the accelerated model performs comparably to the original.

To estimate the generation cost, we define the concept of equivalent steps $K$, expressed as $K = \sum_{k=1}^{K^{\text{step}}}(1 - C_k^{\text{cache}}) \times (1 - C_k^{\text{sparse}})$, where $K^{\text{step}}$ is the total number of steps during generation, determined by $\mathcal{P}^{\text{step}}$. Here, $C_k^{\text{cache}}$ and $C_k^{\text{sparse}}$ represent the cost reductions (normalized into $[0,1]$) achieved by reusing cache and applying sparse attention at the $k$-th step, respectively. In practice, we round $K$ to its nearest integer. The final reward for an image is then expressed as $r_i = \frac{1}{K}\sum_{k=1}^K \lambda^{K-k}(q_i + \omega d_i)$, where $\lambda \in (0, 1)$ is a decay factor that penalizes higher generation costs and $\omega$ denotes the weight of the discriminator in reward.

We can adopt the equation in Section 3.1 to obtain the each sample's advantage $A_i$. Based on the formulation of GRPO, our training process can be formulated as:

$$\mathcal{J}_{\text{GRPO-RAPID}^3} = \frac{1}{G}\sum_{i=1}^G \left( \min\left(\phi_i A_i, \text{clip}\left(\phi_i, 1 - \varepsilon, 1 + \varepsilon\right) A_i\right) \right), \tag{8}$$

where $\phi_i = \frac{\pi_\theta(I_i|c)}{\pi_{\theta_{\text{old}}}(I_i|c)}$. Here, $\pi_\theta(I_i \mid c)$ represents the likelihood of generating the image $I_i$ given the text condition $c$, under the current parameters $\theta$ of the three policy heads, while $\pi_{\theta_{\text{old}}}$ uses the old parameters. The KL divergence term is omitted because the policy heads are randomly initialized, and therefore there is no reference model.

**Training the discriminator model.** Relying solely on the image reward model $Q$ may lead policy heads to exploit or "hack" the reward model, rather than genuinely improving acceleration strategies. To address this, we introduce the discriminator model $D$, a binary classification model designed to distinguish between images generated by the diffusion transformer $G$ with and without acceleration strategies applied. It enhances the reward of accelerated samples from $G$ that remain close to those without acceleration. This makes the discriminator complementary to the reward model, ensuring a more robust training process.

To achieve this, we first allow the generation model $G$, without acceleration, to sample images and construct the positive dataset $\mathcal{I}^{\text{origin}}$. Next, we initialize the policy heads to accelerate $G$, sample images again, and construct the negative dataset $\mathcal{I}^{\text{accele}}$. Both datasets are then employed to train the discriminator. During the reinforcement learning process, the images sampled using GRPO can be used to update the negative dataset $\mathcal{I}^{\text{accele}}$. Cross entropy loss is employed during training. As mentioned in the previous paragraph, we employ the discriminator model $D$ as an additional reward model to provide a supplementary reward signal, which is combined with the reward from the image reward model $Q$ to produce the final reward signal.

This adversarial training process enables joint improvement of the policy models and discriminator. The policy heads are trained to assist $G$ in efficiently producing high-quality images that can fool the discriminator, while the discriminator is optimized to better distinguish between data generated by $G$ with and without acceleration.

## 4 EXPERIMENT

**Model configurations.** We conduct experiments using two diffusion transformers: Stable Diffusion 3 (SD3) (Esser et al., 2024) and its larger counterpart, FLUX (Labs, 2024). For the reward model, we adopt ImageReward (Xu et al., 2023). For the discriminator, we use pre-trained CLIP (Radford et al., 2021) and inset adapters (Chen et al., 2022) for parameter-efficient training. The hyperparameters for the baseline approaches and our method are detailed in Section G and Section H, respectively.

**Datasets and evaluation.** We use 20K prompts randomly sampled from Byeon et al. (2022) and COCO 2017 (Lin et al., 2014) training set to train our policy heads. For evaluation, we use 5,000 images with prompts from the COCO 2017 validation set, along with metrics such as CLIP-Score (Zhengwentai, 2023), and Aesthetic v2 (Schuhmann, 2022) for a comprehensive assessment. Additionally, two comprehensive benchmarks, HPS (Wu et al., 2023) with 1,600 prompts and GenEval Ghosh et al. (2023) with 553 prompts, are introduced to demonstrate the generalization capability of our method. We report the latency measured on an NVIDIA H20 GPU.

### 4.1 RESULTS ON SD3

We compare RAPID[3] with other acceleration techniques, including common step-reduction, two cache reuse methods—$\Delta$-DiT (Chen et al., 2024) and TeaCache (Liu et al., 2024)—and a sparse attention mechanism, SpargeAttn (Zhang et al., 2025a). $\Delta$-DiT employs a uniform interval for all image generations, while TeaCache introduces a manually set threshold to control accumulation errors, enabling a certain degree of adaptivity. For our method, we set the decay factor $\lambda$ in our method is set to 0.97, resulting in around $2.92\times$ acceleration. From Table 1, we observe that RAPID[3] achieves the best balance across all metrics while delivering the highest acceleration ratio compared to its counterparts. This demonstrates that the learned adaptive policies outperform both uniform policies and manually designed adaptive policies, validating the effectiveness of our approach.

Additionally, we compare our method with TPDM (Ye et al., 2024), which only adjusts the generation steps per image. The results show that our method surpasses TPDM across all metrics, validating the superiority of our tri-level acceleration design compared to a single-strategy approach.

Table 1: **Results on SD3 (Esser et al., 2024)**. **Bold** highlights the best results across various acceleration methods. We report latency of the diffusion transformer, excluding the text encoder and VAE decoder. Additional comparison results are provided in Table 8.

| Method | Latency (s) ↓ | Speed ↑ | COCO CLIP ↑ | COCO Aesthetic ↑ | HPS Score ↑ | GenEval Correct ↑ | GenEval Overall ↑ |
|---|---|---|---|---|---|---|---|
| SD3 28-steps | 5.77 | 1.00 × | 32.05 | 5.31 | 28.83 | 67.81 | 69.01 |
| *static or manually designed adaptive acceleration methods* | | | | | | | |
| SD3 9-steps | 1.98 | 2.91 × | 31.88 | 5.21 | 27.67 | 60.58 | 61.67 |
| w/ TeaCache $_{\delta=0.15}$ | 2.20 | 2.62 × | 32.02 | 5.25 | 27.87 | 61.48 | 62.81 |
| w/ $\Delta$-DiT $_{N=4}$ | 3.76 | 1.53 × | 31.91 | 5.12 | 27.67 | 57.69 | 58.67 |
| w/ SpargeAttn | 5.08 | 1.13 × | 31.39 | 5.02 | 27.16 | 43.94 | 45.01 |
| *dynamic acceleration methods* | | | | | | | |
| w/ TPDM | 2.32 | 2.48 × | 31.98 | 5.25 | 27.75 | 59.67 | 60.70 |
| w/ RAPID[3] (Ours) | **1.97** | **2.92** × | **32.09** | **5.26** | **28.07** | **62.57** | **63.48** |

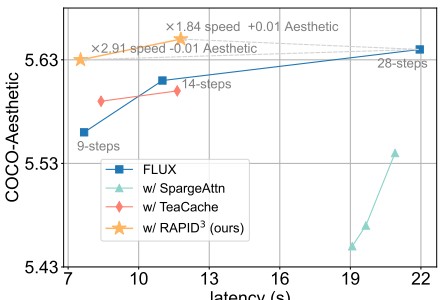

Figure 3: **Latency vs. quality trade-off.**

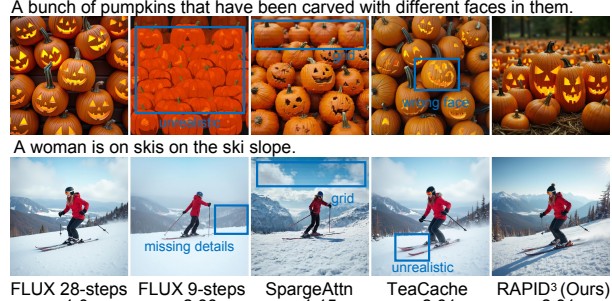

Figure 4: **Visual comparison.** The acceleration ratio relative to the original FLUX 28-steps is reported.

## 4.2 SCALE UP TO FLUX

**Comparison with other acceleration strategies.** In Figure 3, we compare the trade-off between generation latency and quality across various acceleration methods on COCO. We set $\lambda$ in RAPID[3] to 0.97 and 0.90, respectively, to achieve varying acceleration ratios. Both our method and Tea-Cache (Liu et al., 2024), as well as directly reducing steps, significantly accelerate generation speed. However, our method demonstrates greater robustness in maintaining performance, verifying its effectiveness in larger diffusion transformers.

In Figure 4, we compare our approach with other acceleration techniques in terms of visual quality. For this comparison, SpargeAttn, TeaCache, and our method all use the fastest point on the curve from Figure 3. The results demonstrate that our method better preserves visual quality, highlighting the importance of dynamically selecting acceleration strategies.

**Comparison with dynamic model.** DyFLUX (Zhao et al., 2026) extends DyDiT (Zhao et al., 2024) to FLUX (Labs, 2024). It introduces a more sophisticated training process to fine-tune the generator, enabling it to learn an acceleration strategy with per-image adaptivity. As shown in Table 2, our RAPID[3] achieves superior generation quality while also delivering faster inference. Notably, in terms of training cost—both data and computational efficiency—our method outperforms DyFLUX significantly, highlighting its clear superiority.

**Visualization.** In Figure 5, we illustrate the images generated by the original FLUX and our RAPID[3]. To facilitate a clear comparison with the original 28-step generation process, we use the equivalent step as metric, defined in Section 3.4, to represent the generation time cost. For images with a single object and a simple scenario (*e.g.* the image in the first column), our method requires fewer equivalent steps. Conversely, for images with multiple objects and complex scenarios (*e.g.*

Table 2: **Comparison between RAPID³ and DyFLUX.** Compared to DyFLUX (Zhao et al., 2026), our method, significantly improves inference speed, maintains generation quality competitive with the original FLUX, and requires substantially less training cost.

| method | training | | inference | |
| | GPU hours ↓ | data ↓ | latency (s) ↓ | Aesthetic ↑ |
| --- | --- | --- | --- | --- |
| FLUX | - | - | 22.15 | 5.64 |
| DyFLUX | 38,000 | 3M image-text | 13.93 | 5.29 |
| RAPID³ (Ours) | 400 | 20K text only | 8.30 | 5.63 |
| | ≈1% | ≪0.7% | -40% | +0.34 |

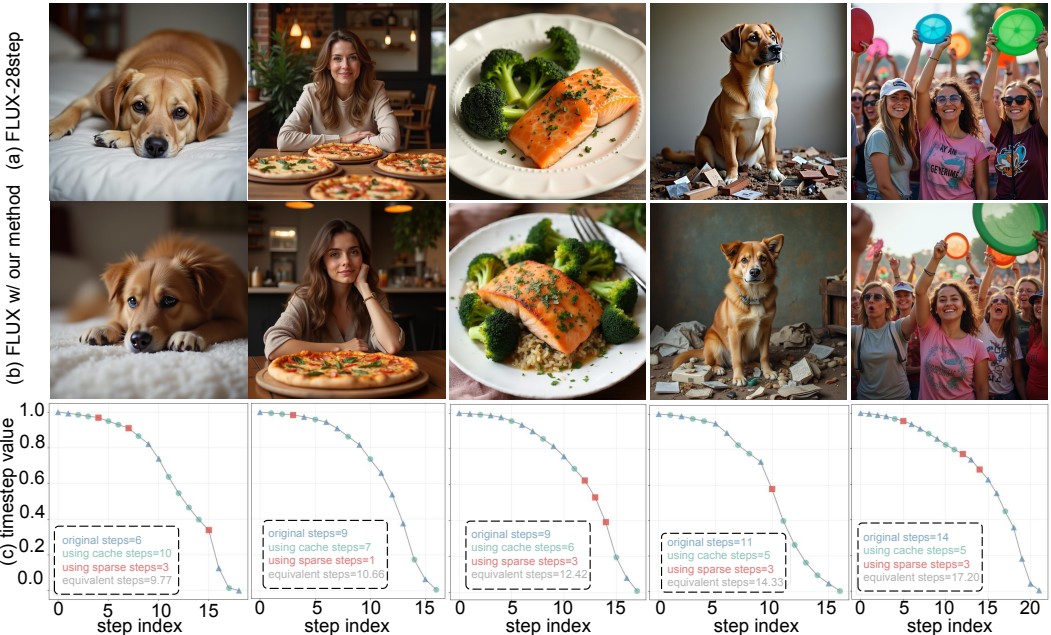

Figure 5: **Visual comparison based on FLUX.** (a) Images generated by the origin FLUX with the default 28 steps. (b) Images generated by FLUX accelerated with RAPID³. (c) The generation processes for (b), where equivalent steps represent the generation time cost for each image.

the image in the last column), it requires more equivalent steps. This demonstrates that our method has learned to adaptively adjust its acceleration strategy based on the complexity of each image. Additional visualizations are provided in Section J.

### 4.3 ANALYSIS

For fairness, we adjust the hyperparameters to ensure similar equivalent steps across all experiments. The color denotes the default setting of RAPID³. All evaluations are conducted on COCO.

**Effectiveness of different candidate acceleration strategies.** In Table 3, we present the results of our method using different candidate acceleration strategies, including step-skip, cache-reuse, and sparse-attention, referred to as "step", "cache", and "sparse", respectively. The results show that progressively incorporating these three acceleration strategies leads to increasingly improved generation performance (*e.g.* CLIP score). Notably, RAPID³ with our default setting, which uses all three strategies, achieves the best average performance. This improvement can be attributed to the expanded solution space provided by more acceleration strategies, enabling RAPID³ to identify a better acceleration approach for each image generation process.

**Effectiveness of the reward model and discriminator model.** We individually remove the reward model $\mathcal{Q}$ and the discriminator $\mathcal{D}$ from the training of our method to evaluate their effectiveness.

Table 3: **Effectiveness of different candidate acceleration strategies.** The symbol ✓ indicates that the corresponding strategy is used in our method.

| step | cache | sparse | CLIP ↑ | Aesthetic ↑ | HPS Score ↑ |
|------|-------|--------|--------|-------------|-------------|
| ✓ | | | 31.98 | 5.25 | 27.75 |
| ✓ | ✓ | | 32.04 | 5.25 | 27.86 |
| | ✓ | ✓ | 31.96 | 5.13 | 27.80 |
| ✓ | ✓ | ✓ | **32.09** | **5.26** | **28.07** |

(header: strategy — step, cache, sparse | COCO — CLIP ↑, Aesthetic ↑ | HPS Score ↑)

Table 4: **Effectiveness of reward model $\mathcal{Q}$ and discriminator model $\mathcal{D}$.** In $\mathcal{Q} + \mathcal{D}$, $\omega$ controlling the weight of the reward from $\mathcal{D}$ is set to 1.0. IR denotes the score from ImageReward (Xu et al., 2023).

| Method | IR | CLIP ↑ | Aesthetic ↑ | HPS Score ↑ |
|--------|-----|--------|-------------|-------------|
| only $\mathcal{Q}$ | **0.9605** | 32.04 | 5.18 | 27.72 |
| only $\mathcal{D}$ | 0.9538 | 31.91 | 5.24 | 27.96 |
| $\mathcal{Q} + \mathcal{D}$ | 0.9574 | **32.09** | **5.26** | **28.07** |

(COCO spans CLIP ↑ and Aesthetic ↑)

Results are presented in Table 4. We additionally include the ImageReward score (IR), used during training, as an additional evaluation metric. When only the reward model $\mathcal{Q}$ is employed, training process degrades to standard reinforcement learning with GRPO, achieving the highest ImageReward score. However, its performance on other metrics drops significantly, as the reward siginal relies on the ImageReward score, making it susceptible to reward hacking. As a result, the policy model focuses solely on optimizing the reward from $\mathcal{Q}$, neglecting the actual generation quality.

In contrast, our method incorporates the discriminator $\mathcal{D}$ into the training process, effectively alleviating the reward hacking problem and maintaining strong performance across various metrics, highlighting the importance of the proposed adversarial reinforcement learning. For completeness, we also conduct an experiment using only the discriminator $\mathcal{D}$ during training and find that it does not outperform our method, further highlighting the importance of integrating both $\mathcal{Q}$ and $\mathcal{D}$.

## 5 CONCLUSION

In this study, we address the challenge of accelerating diffusion transformers in a per-image adaptive manner without modifying their parameters. To achieve this, we introduce RAPID[3]: Tri-Level Reinforced Acceleration Policies for Diffusion Transformer. RAPID[3] employs three lightweight policy heads, optimized via Group Relative Policy Optimization, to select Step-Skip, Cache-Reuse, and Sparse-Attention strategies at each timestep, significantly improving the generation speed. To mitigate reward hacking problem, we incorporate an adversarially learned discriminator to ensure robust policy learning. Extensive experiments demonstrate the effectiveness of RAPID[3], and we anticipate our method will inspire further advancements in accelerating diffusion transformers.

**Limitations and future work.** Our method still relies on training to learn acceleration policies. Incorporating prior knowledge to adaptively select acceleration strategies could further ease the training burden. Additionally, extending the proposed method to video generation models (Zheng et al., 2024; Yang et al., 2024; WanTeam et al., 2025) and editing models (Ma et al., 2025a;b;c; 2023; 2022) warrants further exploration in the future.

**Acknowledgment.** This work was supported by Damo Academy through Damo Academy Research Intern Program. The computational work involved in this research work is also supported by NUS IT's Research Computing group using grant numbers CFP02-CF-004. Yang You's research group is being sponsored by NUS startup grant (Presidential Young Professorship), Singapore MOE Tier-1 grant, ByteDance grant, ARCTIC grant, SMI grant (WBS number: A8001104-00-00), Alibaba grant, and Google grant for TPU usage.

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

## ACKNOWLEDGMENT OF LLM USAGE

All writing, visualizations, and experiments **are completed by the authors**. LLMs (*e.g.*, GPT-4o) are **used solely to refine the writing.**

We organize our appendix as follows:

**Additional details of methods and experiments:**

- Section A: Pipeline of adversarial reinforcement learning in our method.
- Section B: Implementation details for compatibility with classifier-free guidance (CFG).
- Section C: Details of manually combined strategies.
- Section D: Sensitivity to RL method.
- Section E: Comparison with additional state-of-the-art techniques.
- Section F: Impact of training data scale.
- Section L: Experiments on DPG-Bench.
- Section M: Comparison with learning-based cache.

**Experimental settings:**

- Section G: Details of training-free methods in comparison.
- Section H: More implementation details of our method.

**Visualizations**

- Section I: Additional visual comparison with other acceleration techniques.
- Section J: Additional visualization results.
- Section K: Visualization of the distribution patterns in dynamic acceleration strategies.

## A    PIPELINE OF ADVERSARIAL REINFORCEMENT LEARNING

In Algorithm 1, we present the pipeline of the proposed adversarial reinforcement learning in RAPID$^3$.

---

**Algorithm 1:** The pipeline of adversarial reinforcement learning in RAPID$^3$.

---

**Input:** Pre-trained diffusion transformer $\mathcal{G}$ and image reward model $\mathcal{Q}$

1   Randomly initialize the discriminator $\mathcal{D}$ and policy heads $\mathcal{P} = \{\mathcal{P}^{\text{step}}, \mathcal{P}^{\text{cache}}, \mathcal{P}^{\text{sparse}}\}$

2   **while** *training* **do**

3     // Training the discriminator model

4     **If** $\mathcal{I}^{\text{origin}}$ is None **then**: $\mathcal{I}^{\text{origin}} \leftarrow \mathcal{G}$ samplers *w/o* acceleration strategies from $\mathcal{P}$ **end** ;

5     **If** $\mathcal{I}^{\text{accele}}$ is None **then**: $\mathcal{I}^{\text{accele}} \leftarrow \mathcal{G}$ samples *w/* acceleration strategies from $\mathcal{P}$ **end** ;

6     **for** $I \in \mathcal{I}^{\text{origin}} \cup \mathcal{I}^{\text{accele}}$ **do**

7       *update* $\mathcal{D}$ with cross entropy loss ;

8     **end**

9     // Training policies with reinforcement learning

10     **foreach** *training iteration* **do**

11       $\{I_i\}_{i=1}^{G} \leftarrow \mathcal{G}$ conducts sampling *with* acceleration strategies from $\mathcal{P}$;

12       $\{r_i\}_{i=1}^{G} \leftarrow$ obtain rewards of $\{I_i\}_{i=1}^{G}$ based on $\mathcal{Q}$ and $\mathcal{D}$ ;

13       *update* policy models $\mathcal{P}$ with $\mathcal{J}_{\text{GRPO-RAPID}^3}$ ;

14       *update* negative dataset $\mathcal{I}^{\text{accele}}$ with $\{I_i\}_{i=1}^{G}$;

15     **end**

16   **end**

**Output:** Policy heads $\mathcal{P}$ that can select the acceleration strategy for each image generation

---

## B    IMPLEMENTATION DETAILS FOR COMPATIBILITY WITH CLASSIFIER-FREE GUIDANCE

Our method is compatible with Classifier-Free Guidance (CFG) (Ho & Salimans, 2022) and actually all experiments are conducted using CFG. Below, we outline the implementation details:

- SD3 (Esser et al., 2024): In the default settings of SD3, the CFG scale is set to 7.0. During generation with CFG, the batch size is configured to two, consisting of one sample with a textual condition and another with a null condition. The acceleration is dependent on the status of the conditioned sample.

- FLUX (Labs, 2024): As the CFG scale has already been distilled into the FLUX, generation can be conducted directly using our method with a batch size of 1.

## C    COMPARISON WITH MANUALLY COMBINED STRATEGIES

To demonstrate the superiority of the proposed learned policy in our method, we compare it against manually combining different acceleration strategies, as shown in Table 6. The details of manually combined strategies are presented in Table 5. In these methods, we manually integrate reduced sampling steps, feature caching, and sparse attention Specifically, for reducing sampling steps, we adjust the sampling schedule directly. For feature caching, we employ the hand-crafted adaptive method TeaCache (Liu et al., 2024), while using SpargeAttn (Zhang et al., 2025a) as the sparse mechanism.

We observe that the proposed RAPID$^3$ significantly outperforms all manual strategies, demonstrating that simply combining acceleration strategies does not lead to better performance. In fact, the joint introduction of step skipping, cache reuse, and sparse attention greatly expands the search space, making it difficult to manually identify optimal strategies. This often destabilizes the gen-

eration process and leads to unsatisfactory generation quality. This highlights the importance of learnable policies in our method.

Table 5: **Comparison with manual acceleration strategies.**

| Method | Latency (s) | Step | Cache | Sparse |
|--------|-------------|------|-------|--------|
| manual-1 | 3.27 | 21 | $\delta = 0.15$ | $\zeta_1 = 0.05, \zeta_2 = 0.06$ |
| manual-2 | 2.04 | 28 | $\delta = 0.20$ | $\zeta_1 = 0.05, \zeta_2 = 0.06$ |
| manual-3 | 2.38 | 26 | $\delta = 0.12$ | $\zeta_1 = 0.20, \zeta_2 = 0.21$ |

Table 6: **Comparison with manual acceleration strategies.** Our method significantly outperforms them.

| Method | Latency (s) ↓ | COCO CLIP ↑ | COCO Aesthetic ↑ | HPS Score ↑ |
|--------|---------------|-------------|------------------|-------------|
| manual-1 | 3.27 | 31.43 | 4.92 | 27.16 |
| manual-2 | 2.04 | 31.34 | 4.95 | 27.48 |
| manual-3 | 2.38 | 29.82 | 4.85 | 26.94 |
| RAPID[3] | **1.97** | **32.09** | **5.26** | **28.07** |

## D    SENSITIVITY TO RL METHOD

In addition to GRPO, we also evaluate RLOO (Ahmadian et al., 2024), a RL approach that has proven effective in LLM training. The primary difference between GRPO and RLOO lies in how the advantage is obtained. As shown in Table 7, replacing GRPO with RLOO in our method also achieves competitive performance, demonstrating the robustness of our approach across different reinforcement learning approaches.

Table 7: **Replacing the GRPO with RLOO (Ahmadian et al., 2024).** RAPID[3] demonstrates robustness across two RL approaches.

| RL method | COCO CLIP ↑ | COCO Aesthetic ↑ | HPS Score ↑ |
|-----------|-------------|------------------|-------------|
| GRPO | **32.09** | **5.26** | **28.07** |
| RLOO | 32.10 | 5.27 | 28.06 |

## E    ADDITIONAL COMPARISON WITH SOTA METHODS

To supplement Table 1, we incorporate additional methods into our comparison on COCO dataset (Lin et al., 2014), including ToMeSD (Bolya & Hoffman, 2023), AT-EDM (Wang et al., 2024), SDTM (Fang et al., 2025), TokenCache (Lou et al., 2024), DyDiT (Zhao et al., 2024), and ToCa (Zou et al., 2024), as shown in Table 8. The consistent superiority of our method over these state-of-the-art acceleration techniques further underscores its effectiveness and significance.

## F    IMPACT OF TRAINING DATA SCALE

In Table 9, we further investigate the impact of training data scale on the performance of our method. The results show that even with just 5K text-only training samples, our method achieves competitive performance, highlighting its data efficiency. Since our approach keeps the original generator frozen to reduce training costs, the relatively modest improvement observed when increasing the training data to 40K is expected. To balance performance and training efficiency, we adopt 20K samples as the default setting.

## G    DETAILS OF TRAINING-FREE METHODS

**Details of TeaCache.** TeaCache (Liu et al., 2024) is a representative acceleration method leveraging cache reuse. It manually designs a strategy to accelerate inference with per-image adaptivity.

| Method | Acceleration Category | Speed ↑ | CLIP ↑ | Aesthetic ↑ |
|---|---|---|---|---|
| SD3 | - | 1.00 × | 32.05 | 5.31 |
| ToMeSD | Token Merging | 1.50 × | 30.39 | 5.03 |
| AT-EDM | Token Pruning | 1.54 × | 30.27 | 5.02 |
| SDTM | Token Merging | 1.56 × | 31.59 | 5.23 |
| TokenCache | Cache-Reuse | 1.49 × | 31.43 | 5.21 |
| DyDiT | Dynamic Neural Network | 1.57 × | 31.48 | 5.22 |
| ToCa | Cache-Reuse | 2.67 × | 32.05 | 5.24 |
| RAPID[3] (ours) | Dynamic Acceleration | **2.92 ×** | **32.09** | **5.26** |

Table 8: Comparison of methods across different acceleration categories and evaluation metrics.

Table 9: **Impact of training data scale.** The training data consists of text only. We use 20K samples as the default setting to balance performance and training efficiency.

| Method | COCO | | HPS |
|---|---|---|---|
| | CLIP ↑ | Aesthetic ↑ | Score ↑ |
| SD3 | 32.05 | 5.31 | 28.83 |
| 5K | 32.04 | 5.23 | 27.91 |
| 20K | **32.09** | 5.26 | 28.07 |
| 40K | **32.09** | **5.27** | **28.26** |

The method employs a threshold, $\delta$, to decide whether to use the cache or perform computation. Specifically, if the accumulated difference between the latent maps of two consecutive timesteps exceeds the threshold $\delta$, the model performs a full computation and updates the cache. Otherwise, it directly uses the cached residual to skip the model's computation. Larger $\delta$ brings more significant acceleration while also hurts the performance.

For experiments with SD3, we set $\delta = 0.15$. For FLUX, we use two settings, $\delta = 0.15$ and $\delta = 0.25$, to balance the trade-off between latency and generation quality.

**Details of $\Delta$-DiT.** $\Delta$-DiT (Chen et al., 2024) is a representative acceleration method that leverages cache reuse while employing a uniform strategy to accelerate different image generation. The method divides the generation process into two distinct stages. In the first stage, the latter half of the network layers can use cached residuals, while the earlier layers perform full computations. Conversely, in the second stage, the earlier half of the layers can use cached residuals, and the latter layers always conduct full computations.

In the experiment on SD3, for the layers that using caching, the interval for performing computations to update the cache, $N$, is set to 4. However, it is challenging to directly apply this approach to FLUX, which consists of two different types of layers and was not explored in the original paper. Therefore, we conduct our experiments using SD3.

**Details of SpargeAttn.** SpargeAttn (Zhang et al., 2025a) is a representative method for sparsifying the computation of attention. It employs thresholds $\zeta_1$ and $\zeta_2$ to control the difference between the results of attention with and without sparsification. Larger values of $\zeta_1$ and $\zeta_2$ offer better acceleration but inevitably introduce performance degradation.

For experiments with SD3, we set $\zeta_1 = 0.20$ and $\zeta_2 = 0.21$. For experiments with FLUX, we use three settings: $[\zeta_1 = 0.07, \zeta_2 = 0.08]$, $[\zeta_1 = 0.20, \zeta_2 = 0.21]$, and $[\zeta_1 = 0.30, \zeta_2 = 0.31]$.

# H   MORE IMPLEMENTATION DETAILS OF RAPID[3]

In Table 10, we present the default implementation details of our approach.

Table 10: **Default Implementation Details of RAPID[3].**

|  |  |
|---|---|
| *details of training* | |
| device | $8\times$ NVIDIA H20 GPU |
| total batch size | 128 |
| learning rate | 1e-5 |
| weight decay | 0.1 |
| optimizer | AdamW |
| samples per group in GRPO | 4 |
| *details of reward* | |
| decay factor $\lambda$ | 0.97 |
| weight of the discriminator in reward $\omega$ | 1.0 |
| $C_k^{\text{cache}}$ | 0.95 (measured) |
| candidate sparse attention | $[\zeta_1 = 0.07, \zeta_2 = 0.08],$ $[\zeta_1 = 0.10, \zeta_2 = 0.11],$ $[\zeta_1 = 0.20, \zeta_2 = 0.21]$ |
| $C_k^{\text{sparse}}$ | 0.05, 0.07, 0.10 (measured) |

## I ADDITIONAL VISUAL COMPARISON WITH OTHER ACCELERATION TECHNIQUES

In Figure 6, we provide additional visual comparisons with other acceleration techniques on FLUX (Labs, 2024).

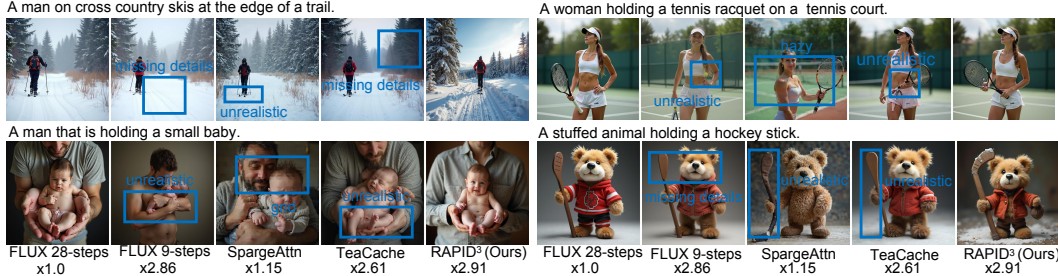

Figure 6: **Additional visual comparison with other acceleration techniques.**

## J ADDITIONAL VISUALIZATION RESULTS

In Figures 7 and 8, we present additional visualizations of images generated by the original FLUX (Labs, 2024) and its accelerated counterpart using our method. The results demonstrate that our method preserves visual quality more effectively while achieving the best acceleration ratio, verifying the importance of dynamically selecting acceleration strategies.

Moreover, we can observe several notable trends emerging from our dynamic acceleration strategies:

- Step-Skip Stride: The stride of step-skip is relatively small when $t$ approaches 1.0 (near the noise distribution) but becomes larger as $t$ approaches 0.0 (near the image distribution). This behavior can be attributed to the model's requirements during different stages of generation. At the initial stage, the model focuses on generating the overall shape, structure, and composition of the image, which has a significant impact on the final image quality. Therefore, our method employs finer-grained steps during this phase to ensure higher image quality.

- Cache-Reuse: Cache reuse is typically applied at intervals of 0–4 steps, as the computations in some consecutive steps are similar and can be replaced by cached results from previous steps.

- Sparse-Attention: Sparse attention is applied in an image-dependent manner, meaning the frequency of its usage varies depending on the image. For instance, as shown in Figure 7, 5 steps of sparse attention are used for the rightmost image, while only 1 step is applied for the middle image.

These results, along with the ablation study presented in Table 3, highlight that the three acceleration policies work synergistically and effectively complement each other.

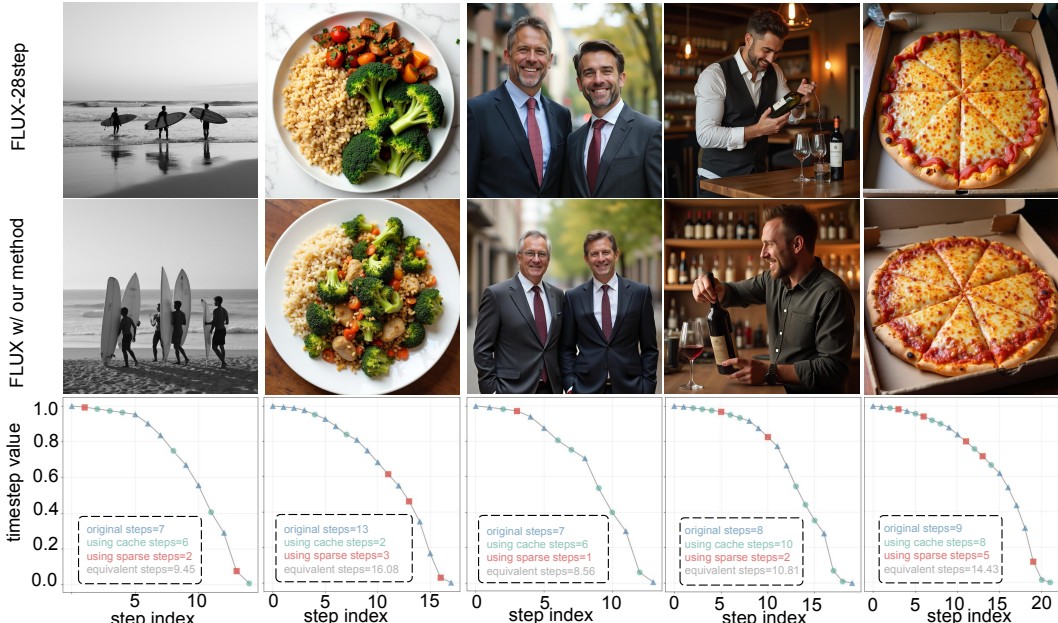

Figure 7: **Additional visualizations of images generated by the original FLUX (Labs, 2024) and its accelerated version using our method. (1)**

## K    DISTRIBUTION PATTERNS IN DYNAMIC ACCELERATION STRATEGIES

In Figure 9, we present the distribution patterns of dynamic acceleration strategies learned by our policy heads, derived from 5,000 samples of the COCO dataset (Lin et al., 2014). Specifically, Figure 9(a) illustrates the distribution of total steps used during generation, while Figures 9(b) and 9(c) demonstrate the distributions of steps involving the use of cache and sparse attention, respectively. These findings further confirms the ability of our method to adaptively select acceleration strategies for each image generation process, leading to a diverse range of strategies.

## L    EXPERIMENTS ON DPG-BENCH

We conduct experiments on a comprehensive benchmark, DPG-Bench (Hu et al., 2024), to evaluate the proposed method against various baselines. The results, presented in Table 11, demonstrate that the proposed dynamic acceleration strategy remains robust even in such challenging settings.

## M    COMPARISON WITH LEARNING-BASED CACHE

Recent advancements in feature-caching for accelerating diffusion transformers, such as HarmoniCa (Huang et al., 2024) and Learning-to-Cache (Ma et al., 2024a), have started incorporating

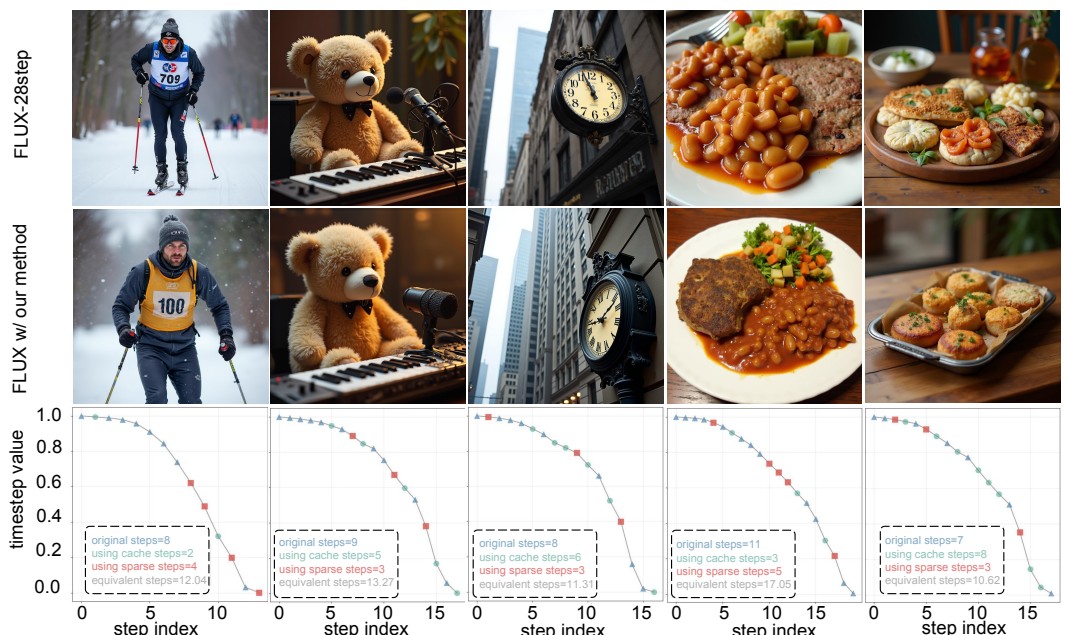

Figure 8: **Additional visualizations of images generated by the original FLUX (Labs, 2024) and its accelerated version using our method. (2)**

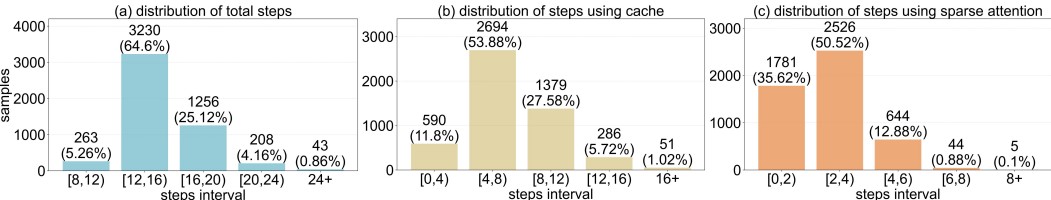

Figure 9: **Visualization of distribution patterns in dynamic acceleration strategies.** In (a), we visualize the distribution of total steps for sample generation. In (b) and (c), we demonstrate the distribution of steps using cache and sparse attention. For instance, 3,230 samples are generated using 12–16 steps, 2,694 samples utilize cache reuse with 4–8 steps, and 2,526 samples employ sparse attention with 2–4 steps during the generation process. This experiment is based on our model, which achieves a 2.91 × speedup over FLUX (Labs, 2024), as illustrated in Figure 3.

Table 11: Experiments on DPG-Bench (Hu et al., 2024).

| Method | Global | Entity | Attribute | Relation | Other | Average |
|---|---|---|---|---|---|---|
| SD3 28-steps | 88.09 | 87.81 | 85.79 | 86.16 | 87.00 | 81.29 |
| SD3 9-steps | 85.78 | 85.38 | 82.12 | 83.72 | 83.70 | 76.77 |
| w/ $\Delta$-DiT $_{N=4}$ | 68.61 | 73.69 | 76.46 | 79.78 | 75.69 | 64.58 |
| w/ TeaCache | 84.03 | 85.56 | 84.15 | 88.46 | 83.20 | 77.53 |
| w/ TPDM | 80.00 | 78.35 | 80.26 | 81.76 | 79.19 | 72.46 |
| w/ SpargeAttn | 80.39 | 81.66 | 81.53 | 79.48 | 79.66 | 73.73 |
| w/ RAPID[3] (Ours) | 88.94 | 84.91 | 84.36 | 86.30 | 85.77 | 78.81 |

learning-based processes. These methods typically introduce routers to decide whether to perform computation for a block or reuse the cache from a previous step. To compare our proposed method with such designs, we conduct experiments on HarmoniCa, an improved version of Learning-to-Cache, with results presented in Table 12. The results demonstrate that our method outperforms the

Table 12: **Comparison with HarmoniCa (Huang et al., 2024)**.

| Method | Speed ↑ | COCO | | HPS | GenEval | |
| --- | --- | --- | --- | --- | --- | --- |
| | | CLIP ↑ | Aesthetic ↑ | Score ↑ | Correct ↑ | Overall ↑ |
| SD3 28-steps | 1.00 × | 32.05 | 5.31 | 28.83 | 67.81 | 69.01 |
| w/ HarmoniCa | 2.01 × | 31.75 | 5.23 | 28.05 | 59.67 | 60.99 |
| w/ RAPID³ (Ours) | **2.92 ×** | **32.09** | **5.26** | **28.07** | **62.57** | **63.48** |

learning-based feature-caching approach, HarmoniCa, underscoring the superiority of our approach. This can be attributed to three key reasons:

- Three-level acceleration strategies: Our method incorporates three levels of acceleration strategies, step-skip, cache-reuse, and sparse-attention, rather than relying solely on cache reuse. This diverse set of strategies offers greater flexibility and optimization potential compared to caching-based methods like HarmoniCa.

- Unconstrained performance upper bound: Traditional learning-based caching paradigms like HarmoniCa are designed to align the performance with the original model, which inherently limits their upper bound to the performance of the original model. In contrast, our method, trained with reinforcement learning, is not bound by this constraint. Instead, it continuously optimizes the model to achieve higher rewards, unlocking the potential for superior performance.

- Its router determines whether to use caching solely based on the timestep, independent of each image generation. This results in inferior performance compared to our image-adaptive accretion strategy.

We hope our design will inspire future work in feature-caching methods to fully realize their combined potential.

