# OpenReview forum: "RAPID$^3$: Tri-Level Reinforced Acceleration Policies for Diffusion Transformer"
_ICLR.cc/2026/Conference — ICLR 2026 Poster_

### Official Review · Reviewer_ZdkW · 2025-10-30

**Soundness:** 2
**Presentation:** 2
**Contribution:** 2
**Rating:** 4
**Confidence:** 4

**Summary:**

This paper introduces RAPID3, a framework that enables per-image adaptive acceleration for large DiT models such as Stable Diffusion 3 and FLUX, without modifying the base generator’s parameters. RAPID3 formulates the sampling process of a DiT as a Markov Decision Process (MDP), in which each denoising step corresponds to a state transition. It attaches three lightweight policy heads, each trained via RL, to dynamically determine Step-Skip Policy, Cache-Reuse Policy, Sparse-Attention Policy. Empirically, RAPID3 achieves nearly 3× faster sampling on SD3 and FLUX with minimal image fidelity degradation.

**Strengths:**

RAPID3 reframes the inference-time efficiency problem of diffusion transformers as a reinforcement learning (RL) control problem, treating each denoising step as a state-action transition. This perspective is both conceptually elegant and practically impactful. The proposed tri-level policy design—encompassing Step-Skip, Cache-Reuse, and Sparse-Attention strategies—is a novel unification of previously disparate acceleration techniques under a single learnable framework.

**Weaknesses:**

1. Limited scope of evaluation metrics and qualitative diversity.

The paper’s evaluation primarily relies on image-level alignment metrics such as CLIP-Score, Aesthetic, and GenEval, which mainly capture prompt consistency or aesthetic preference. However, it overlooks popular benchmarks that directly assess the intrinsic quality and compositional correctness of generated images, such as DPG-Bench or other perception-grounded generation metrics. These benchmarks are designed to measure fidelity, entity consistency, and spatial relations—aspects that are particularly sensitive to dynamic acceleration strategies like cache reuse and sparse attention.

2. Limited exploration of policy coordination and interdependence

While RAPID3 introduces three distinct policy heads (Step-Skip, Cache-Reuse, Sparse-Attention), each is trained independently under a shared reward function. The paper does not fully analyze the inter-policy dependencies—for example, how aggressive step-skipping interacts with sparse attention or cache reuse. In practice, these actions may have compounding or conflicting effects on visual fidelity and stability. A more detailed study on joint versus decoupled optimization, or a hierarchical policy controller that adaptively balances the three heads, would strengthen the understanding of policy synergy and improve robustness.

**Questions:**

Lack of comparison with learned feature caching approaches

The experimental comparisons mainly include training-free baselines (e.g., TeaCache, SpargeAttn) and dynamic fine-tuned models (DyFLUX, TPDM), but they omit recent learned feature caching methods such as HarmoniCa [1] and Learning-to-Cache [2], which also learn adaptive caching or reuse strategies through training. These approaches represent a closer baseline to RAPID3, as they similarly attempt to learn data-dependent cache update schedules rather than rely on manual thresholds. Without direct comparison or discussion, it remains unclear how RAPID3’s reinforcement learning–based policy differs from or improves upon these learned caching paradigms in terms of adaptivity, efficiency, and quality preservation. Including such baselines would better position RAPID3 within the broader family of learned, model-aware acceleration methods.

[1] HarmoniCa: HarmonizingTraining and Inference for Better Feature Caching in Diffusion Transformer Acceleration, ICML 2025.

[2] Learning to-cache: Accelerating diffusion transformer via layer caching, NeurIPS 2024.

---

> ### Author Response · Authors · 2025-11-26
> **Reviewer ZdkW 1/2**
>
> >W1: Limited scope of evaluation metrics and qualitative diversity.
> The paper’s evaluation primarily relies on image-level alignment metrics such as CLIP-Score, Aesthetic, and GenEval, which mainly capture prompt consistency or aesthetic preference. However, it overlooks popular benchmarks that directly assess the intrinsic quality and compositional correctness of generated images, such as DPG-Bench or other perception-grounded generation metrics. These benchmarks are designed to measure fidelity, entity consistency, and spatial relations—aspects that are particularly sensitive to dynamic acceleration strategies like cache reuse and sparse attention.
>
>
>
> We sincerely appreciate this valuable suggestion. In response, we conduct experiments on a comprehensive benchmark, DPG-Bench [1], to compare the proposed method against other baselines. Results are presented below:
>
> **Results:**
>
>
> | Method              | Global       | Entity       | Attribute    | Relation     | Other        | Average      |
> |-------------------|--------------|--------------|--------------|--------------|--------------|--------------|
> | SD3 28-steps      | 88.09  | 87.81  | 85.79   | 86.16  | 87.00  | 81.29  |
> | SD3 9-steps       | 85.78  | 85.38  | 82.12  | 83.72  | 83.70  | 76.77  |
> | w/ Delta-Cache        | 68.61  | 73.69  | 76.46 | 79.78 | 75.69  | 64.58  |
> | w/ TeaCache          |  84.03        |  85.56            |  84.15            |  88.46            |     83.20         |      77.53        |
> | w/ TPDM              | 80.00           | 78.35 | 80.26  | 81.76        | 79.19  | 72.46  |
> |  SpargeAttn            | 80.39 | 81.66  | 81.53  | 79.48  | 79.66 | 73.73  |
> | RAPID3 (ours)      | 88.94  | 84.91 | 84.36  | 86.30  | 85.77  | 78.81  |
>
>
> **Conclusion:**
>
> The results demonstrate that the proposed dynamic acceleration strategy achieves the highest average performance, highlighting its robustness in this challenging setting. These findings and analyses have been updated in **Section L of the Appendix** to inspire future work.
>
>
>
>
>
> **Reference:**
>
> [1] ELLA: Equip Diffusion Models with LLM for Enhanced Semantic Alignment, ArXiv 2024.
>
>
>
>
>
>
> >W2: Limited exploration of policy coordination and interdependence.
> While RAPID3 introduces three distinct policy heads (Step-Skip, Cache-Reuse, Sparse-Attention), each is trained independently under a shared reward function. The paper does not fully analyze the inter-policy dependencies—for example, how aggressive step-skipping interacts with sparse attention or cache reuse. In practice, these actions may have compounding or conflicting effects on visual fidelity and stability. A more detailed study on joint versus decoupled optimization, or a hierarchical policy controller that adaptively balances the three heads, would strengthen the understanding of policy synergy and improve robustness.
>
>
> Thank you for this suggestion. Below, we provide our analysis of the learned policy in RAPID3:
>
>
>
> + **Policy coordination:**  From **Figure 5**, we observe the following trends when jointly using the three policies:
>
>     + Step-Skip: The stride of step-skip is relatively **small** when **t approaches 1.0** (near the noise distribution) but **becomes larger** as **t approaches 0.0** (near the image distribution). This behavior can be attributed to the model's requirements during different stages of generation. At the initial stage, the model focuses on generating the overall shape, structure, and composition of the image, which has a significant impact on the final image quality. Therefore, our method employs finer-grained steps during this phase to ensure higher image quality.
>
>     + Cache-Reuse: Cache reuse is typically **applied at intervals of 0–4 steps**, as the computations in **some consecutive steps are similar** and can be replaced by cached results from previous steps.
>
>     + Sparse-Attention: Sparse attention is applied in an **image-dependent** manner, meaning the frequency of its usage varies depending on the image. For example, in our method, 3 steps of sparse attention are used for the leftmost image, while only 1 step of sparse attention is applied for the second-rightmost image.
>
>
>
> + **Policy interdependence:** In **Table 3**, we present ablation studies analyzing the interaction between the different policies. These results demonstrate that the three acceleration policies **work synergistically and effectively complement each other.**.
>
>
> + **Policy optimizaton:** The analyses above demonstrate that our method does **not rely on complex optimization processes** and is capable of effectively learning to **balance different acceleration strategies**. We sincerely appreciate the valuable suggestion from the reviewer, which has inspired us to explore more effective optimization strategies for training policy heads and adaptively balancing them to further enhance performance in future work.
>
>
>
> All the analyses are included in **Section J of the Appendix**.

---

> ### Author Response · Authors · 2025-11-26
> **To Reviewer ZdkW 2/2**
>
> >W3: Lack of comparison with learned feature caching approaches. The experimental comparisons mainly include training-free baselines (e.g., TeaCache, SpargeAttn) and dynamic fine-tuned models (DyFLUX, TPDM), but they omit recent learned feature caching methods such as HarmoniCa [1] and Learning-to-Cache [2], which also learn adaptive caching or reuse strategies through training. These approaches represent a closer baseline to RAPID3, as they similarly attempt to learn data-dependent cache update schedules rather than rely on manual thresholds. Without direct comparison or discussion, it remains unclear how RAPID3’s reinforcement learning–based policy differs from or improves upon these learned caching paradigms in terms of adaptivity, efficiency, and quality preservation. Including such baselines would better position RAPID3 within the broader family of learned, model-aware acceleration methods.
>
>
>
> We sincerely appreciate this insightful suggestion. We conduct experiments to compare with a **representative** learning-based feature-cacing approach, HarmoniCa [1], which is improved upon Learning-to-Cache [2].
>
>
> **Results:**
>
> | Model                     |Speed ↑ | COCO CLIP↑ | COCO Aesthetic↑ | HPS Score↑ | GenEval Correct↑ | GenEval Overall↑ |
> |---------------------------|--------------|-----------|----------------|-----------|-----------------|-----------------|
> | HarmoniCa     | 2.01x        | 31.75     | 5.23           |     28.05      | 59.67           | 60.99           |
> | RAPID                     | 2.92x        | 32.09     | 5.26           | 28.07     | 62.57           | 63.48           |
>
>
>
>
>
> **Analysis:**
>
>
> The results above demonstrate that our method outperforms the learning-based feature caching approach, HarmoniCa, under an aggressive acceleration ratio (> 2.00x). This advantage can be attributed to three key reasons:
>
> + 1. Three-level acceleration strategies: Our method incorporates three levels of acceleration strategies, step-skip, cache-reuse, and sparse-attention, **rather than relying solely on cache reuse**. This diverse set of strategies offers **greater flexibility and optimization potential** compared to caching-based methods like HarmoniCa.
>
> + 2. Unconstrained performance upper bound: Traditional learning-based caching paradigms like  HarmoniCa are designed to **align the performance with the original model**, which inherently **limits their upper bound** to the performance of the original model. In contrast, our method, trained with reinforcement learning, is **not bound** by this constraint. Instead, it continuously optimizes the model to **achieve higher rewards**, unlocking the potential for superior performance.
>
> + 3. Its router determines whether to use caching **solely based on the timestep**, independent of each image generation. This results in inferior performance compared to our image-adaptive accretion strategy.
>
>
> Additionally, we would like to clarify that our method aims to **explore a new paradigm** for accelerating diffusion transformers, whereas methods like HarmoniCa focus on **fully exploiting the potential of caching-based acceleration methods**. This distinction highlights the **orthogonal** nature of these approaches, suggesting their **potential complementarity**.
>
>
> Thank you for this valuable suggestion. We **cite both HarmoniCa[1] and Leaning-to-Cache[2] in the main paper** and include these results, analyses, and a dicussion of these two methods in the **Section M of the Appendix** to facilitate future research.

---

### Official Review · Reviewer_6idy · 2025-11-01

**Soundness:** 3
**Presentation:** 3
**Contribution:** 3
**Rating:** 6
**Confidence:** 3

**Summary:**

This paper presents RAPID3, a reinforcement-learning-based acceleration framework for Diffusion Transformers (DiTs). The key idea is to view the diffusion sampling process as a Markov Decision Process (MDP) and attach three lightweight policy heads—Step-Skip, Cache-Reuse, and Sparse-Attention—in a clean and neat design.

**Strengths:**

1. The story of this paper is reasonable, and the design is clever.

2. The proposed method does not modify or fine-tune the base generator. Training only the small policy heads with GRPO is resource-efficient, which makes the framework practical even for closed-source large models.

**Weaknesses:**

1. It is a little bit weird that the main experiments in Table 1 choose 28 steps as the baseline, which is not a commonly used setting. I want to know why.

2. Given that the method is resource-efficient, more validations on video diffusion or image editing can further strengthen the impact of this paper.

3. The ablation studies are solid but remain surface-level. For example, while the tri-level design (Step/Cache/Sparse) shows quantitative gains, there is little insight into *when* and *why* each policy activates.

**Questions:**

See Weakness

---

> ### Author Response · Authors · 2025-11-26
> **To Reviewer 6idy**
>
> > W1: It is a little bit weird that the main experiments in Table 1 choose 28 steps as the baseline, which is not a commonly used setting. I want to know why.
>
> Thank you. We would like to clarify that we chose 28 steps as it aligns with **the default setting** and is a **common practice in previous works** for acceleration. Details are provided below:
>
> + Both SD3 and FLUX adopt 28 steps as the default setting in their **official implementations** (see [1, 2]).
>
> +  Many prior works, such as [3, 4, 5], focusing on accelerating SD3 or FLUX, primarily conduct experiments with 28 steps.
>
> Furthermore, our method is **agnostic to the total number of sampling steps** and can be applied to accelerate baselines with more sampling steps, such as 50 steps.
>
>
>
>
> **Reference:**
>
> [1] Hugging Face, "Pipeline Stable Diffusion 3 Implementation". Available: https://github.com/huggingface/diffusers/blob/6f1042e36cd588a7b66498f45c3bb7085e4fa395/src/diffusers/pipelines/stable_diffusion_3/pipeline_stable_diffusion_3.py#L781.
>
> [2] Hugging Face, "Pipeline Flux Implementation". Available: https://github.com/huggingface/diffusers/blob/6f1042e36cd588a7b66498f45c3bb7085e4fa395/src/diffusers/pipelines/flux/pipeline_flux.py#L663.
>
> [3] Attend to Not Attended: Structure-then-Detail Token Merging for Post-training DiT Acceleration, CVPR 2025.
>
> [4] DyDiT++: Dynamic Diffusion Transformers for
> Efficient Visual Generation, ArXiv 2025.
>
> [5] Schedule on the fly: Diffusion time prediction for faster and better image generation, CVPR 2025
>
>
>
>
> >W2: Given that the method is resource-efficient, more validations on video diffusion or image editing can further strengthen the impact of this paper.
>
>
> We sincerely appreciate this insightful and constructive suggestion.
>
> + We would like to clarify that the primary objective of this work is to **propose a novel dynamic acceleration paradigm**. As such, our focus has been on the **foundational** image generation models to establish and **validate the core methodology**.
>
>
> + Additionally, in the "Limitations and Future Work" section (**Section 5**), we discusse that extending our method to video generation models or image editing models is indeed a **promising direction** for future research.
>
>
> Inspired by your valuable feedback, we are conducting a preliminary exploration in this direction. However, due to unforeseen changes in computational resources on our end, this work is still ongoing. We are committed to continuing this effort and will publicly release the results once they are ready. Thank you again for your thoughtful input, which has further encouraged us to push the boundaries of our research.
>
>
>
>
>
>
> >W3: The ablation studies are solid but remain surface-level. For example, while the tri-level design (Step/Cache/Sparse) shows quantitative gains, there is little insight into when and why each policy activates.
>
> We sincerely appreciate this insightful question. To clarify, since the policy is learning-based, there is no explicit regulation to activate each acceleration strategy. However, from **Figure 5**, we can observe the following trends:
>
> + Step-Skip: The stride of step-skip is relatively **small** when **t approaches 1.0** (near the noise distribution) but **becomes larger** as **t approaches 0.0** (near the image distribution). This behavior can be attributed to the model's requirements during different stages of generation. At the initial stage, the model focuses on generating the overall shape, structure, and composition of the image, which has a significant impact on the final image quality. Therefore, our method employs finer-grained steps during this phase to ensure higher image quality.
>
> + Cache-Reuse: Cache reuse is typically **applied at intervals of 0–4 steps**, as the computations in **some consecutive steps are similar** and can be replaced by cached results from previous steps.
>
> + Sparse-Attention: Sparse attention is applied in an **image-dependent** manner, meaning the frequency of its usage varies depending on the image. For example, in our method, 3 steps of sparse attention are used for the leftmost image, while only 1 step of sparse attention is applied for the second-rightmost image.
>
> In response to this valuable question, this explanation has been included in **Section J of the Appendix** to provide further insights into the learned dynamic activation of acceleration strategies.

---

### Official Review · Reviewer_Wtnj · 2025-11-01

**Soundness:** 3
**Presentation:** 3
**Contribution:** 3
**Rating:** 6
**Confidence:** 4

**Summary:**

This paper proposes RAPID3, a system that attaches three tiny, trainable policy heads to a frozen Diffusion Transformer (DiT) to adaptively accelerate inference per image without fine-tuning the generator. The three policies are:

- Step-Skip — learns per-timestep jumps (a Beta-parameterized jump).

- Cache-Reuse — decides whether to reuse cached residuals vs recompute.

- Sparse-Attention — chooses among discrete attention sparsity patterns per timestep.


Policies are trained with a group based policy-optimization (GRPO) objective and an adversarial discriminator is used alongside an image reward model to reduce reward-hacking. Experiments on Stable Diffusion 3 and FLUX show ≈3× speedups while retaining competitive metrics (CLIP, Aesthetic, GenEval/HPS) and much lower training cost than dynamic fine-tuning approaches.

**Strengths:**

1. **Practical, low-cost adaptivity.**

    The paper addresses a highly practical problem (fast inference for large DiTs) and presents a solution that keeps the generator frozen, adding only lightweight heads (~0.025% of generator params) — an attractive tradeoff for deployment and proprietary models. The claimed GPU-hour savings vs fully fine-tuned dynamic models is strong evidence of real utility.

2. **Tri-level design is well motivated and complementary.**

    Combining step skipping, cache reuse, and sparse attention covers orthogonal sources of compute (temporal, recomputation, attention cost). Ablations show incremental gains when adding strategies, supporting the architectural choice.

3. **Careful training objective to mitigate reward hacking.**

    The use of an adversarial discriminator alongside an image-reward model is an effective and conceptually clean way to prevent policies from exploiting an automatic metric. The Algorithm and RL setup (GRPO) are reasonable for the deterministic frozen-model environment.

4. **Empirical evidence on strong backbones.**

    Results on SD3 and the larger FLUX backbone are provided, including comparisons to training-free baselines (TeaCache, ∆-DiT, SpargeAttn), single-strategy RL (TPDM), and a dynamic fine-tuned baseline (DyFLUX). Quantitative results and visualization suggest RAPID3 attains large speedups with competitive quality.

5. **Thorough ablations and implementation detail.**

    The appendix and tables include implementation defaults, sensitivity to RL method (GRPO vs RLOO), and data-scale experiments, which strengthen reproducibility and trust in robustness.

**Weaknesses:**

1. **Benchmarks and baselines — potential fairness concerns.**

    The paper compares to many baselines, but details matter: how were hyperparameters for baselines tuned (equivalent computational budget / equivalent “equivalent steps”)? The paper claims equivalent-step fairness but could be clearer on whether baselines were given the same effort for tuning (especially manual adaptive methods). Without fully transparent tuning procedures and seeds, it’s hard to judge whether gains stem from method novelty or hyperparameter selection.

2. **Computational cost during deployment & overheads.**

    Although policy heads are tiny, the per-step decision logic, discriminator CPA during training, and any additional memory for caches add overhead. The latency numbers report the DiT forward pass only (excluding text encoder and VAE decoder); reporting wall-clock end-to-end costs (including policy inference) would give a clearer picture of real speedups. The paper should more explicitly quantify policy inference cost and memory overhead.

**Questions:**

1. **Policy overhead & reproducibility:** Please report the exact inference overhead of running the three policy heads (ms per step) and memory used for caches, and confirm whether the reported latency numbers include policy compute. Also, can you release seeds/hyperparameter configs used for baselines so comparisons are fully reproducible?

2. **Ablation on discriminator strength & dataset bias:** How sensitive are results to the discriminator model choice (architecture, adapter size) and to the reward weight ω? Table 4 shows some effect, but a sweep and discussion of potential adversarial equilibria (where policies game D instead of Q) would be helpful.

3. **Generality to other samplers and guidance strengths:** Does RAPID3 interact with different samplers (DPM-solver variants) and classifier-free guidance scales? Appendix claims compatibility, but please show a small table demonstrating sensitivity to sampler/guidance variations.

---

> ### Author Response · Authors · 2025-11-26
> **To Reviewer Wtnj  1/2**
>
> > W1: Benchmarks and baselines — potential fairness concerns. The paper compares to many baselines, but details matter: how were hyperparameters for baselines tuned (equivalent computational budget / equivalent “equivalent steps”)? The paper claims equivalent-step fairness but could be clearer on whether baselines were given the same effort for tuning (especially manual adaptive methods). Without fully transparent tuning procedures and seeds, it’s hard to judge whether gains stem from method novelty or hyperparameter selection.
>
>
>
> We sincerely appreciate this thoughtful question. We would like to clarify that the hyperparameters for all baselines in Table 1 (for SD3) and Figure 3 (for FLUX) **have been carefully tuned**.
>
> **Explanation:**
> + We adjust the hyperparameters to tune the "equivalent step" of the baselines, allowing them to achieve **meaningful acceleration** while maintaining **reasonable performance**.
> + Detailed explanations of the hyperparameters of each baseline method **are provided in Section G** of the appendix.
>
> Therefore, **the comparison is fair**, and the observed performance improvements are **attributed to our design** rather than solely to hyperparameter tuning.
>
>
> In response to this valuable suggestion, we have **added a note in Lines 356–358** to guide readers to the relevant details.
>
>
>
>
> > W2: Computational cost during deployment & overheads. Although policy heads are tiny, the per-step decision logic, discriminator CPA during training, and any additional memory for caches add overhead. The latency numbers report the DiT forward pass only (excluding text encoder and VAE decoder); reporting wall-clock end-to-end costs (including policy inference) would give a clearer picture of real speedups. The paper should more explicitly quantify policy inference cost and memory overhead.
>
>
> Thanks. Below is the explanation:
>
> + Latency: The latency value includes the wall-clock time for the diffusion transformer, **encompassing both the original generation process and the newly introduced policy heads**. Hence, both Table 1 and Figure 3 demonstrate the **real speedup**.
>
>
>
>
> + Memory: The memory cost introduced by the policy heads is **negligible** (<0.1%) as they are lightweight, and their parameters are **minimal compared to the original generator**. Furthermore, the caching-related memory cost is comparable to TeaCache, requiring just about 0.026GB per image on FLUX in Float16 precision.
>
>
> Therefore, our method is both **cost-effective and practical** for deployment, as also highlighted by Reviewer 6idy and Reviewer ZdkW.
>
>
>
> > Q1: Policy overhead & reproducibility: Please report the exact inference overhead of running the three policy heads (ms per step) and memory used for caches, and confirm whether the reported latency numbers include policy compute. Also, can you release seeds/hyperparameter configs used for baselines so comparisons are fully reproducible?
>
> Thanks. We provide the explanation below:
>
> + The overhead of running three policy heads is **only 2.2ms per step**, accounting for **less than 1%** of the total step latency. The memory cost associated with caching is comparable to TeaCache, requiring just about **0.026GB per image** on FLUX in Float16 precision, as mentioned in our response to Reviewer W2.
>
> + We confirm that the reported latency numbers **include the policy computation**, as clarified in our response to W2.
>
> + The random seed is **set to 0 for all experiments**, and the hyperparameter configurations are detailed in **Section G** of the appendix to ensure reproducibility.

---

> ### Author Response · Authors · 2025-11-26
> **To Reviewer Wtnj  2/2**
>
> > Q2: Ablation on discriminator strength & dataset bias: How sensitive are results to the discriminator model choice (architecture, adapter size) and to the reward weight ω? Table 4 shows some effect, but a sweep and discussion of potential adversarial equilibria (where policies game D instead of Q) would be helpful.
>
> We sincerely appreciate these valuable suggestions. The sensitively analysis are presented below:
>
>
> + Discriminator model choice (architecture, adapter size): By default, we adopt the ViT-B architecture for the discriminator. While we also experiment with a **larger architecture**, ViT-L, we do **not observe significant improvement**. The adapter size (rank) is set to 128 by default. Results on COCO presented below demonstrate that our method is **robust to changes in the rank**, whether it is increased or decreased.
>
>     | r       |CLIP ↑     |Aesthetic ↑|
>     |--------|------------|-----------|
>     |     64 |       32.07     |        5.25   |
>     | 128 (default)|   32.09   |    5.26       |
>     |    256 |       32.09     |    5.26       |
>
>     These results indicate that our method is **not sensitive** to the discriminator model's hyperparameters.
>
>
> + Reward weight w: This weight, set to a default value of 1.0, is used to balance the influence of the discriminator model (D) on the final reward signal. Our experiments on COCO show that setting this weight **too low or too high leads to a performance drop**. Therefore, we **retain 1.0** as it provides **the best balance**.
>
>     | w       |CLIP ↑     |Aesthetic ↑|
>     |---------|-----------|-----------|
>     |     0.1 |    32.05       |  5.19         |
>     | 1.0 (default) |  32.09   |  5.26         |
>     |    10 |    31.89         |  5.23      |
>
>
>
> Additionally, the results of the policies game D are **presented as "only D" in Table 4**, highlighting the importance of jointly using D and Q.
>
>
>
>
> > Q3: Generality to other samplers and guidance strengths: Does RAPID3 interact with different samplers (DPM-solver variants) and classifier-free guidance scales? Appendix claims compatibility, but please show a small table demonstrating sensitivity to sampler/guidance variations.
>
>
> Thanks. Our RAPID3 is **agnostic** to different samplers and classifier-free guidance scales. In this work, both SD3 and FLUX adopt the Euler sampler, consistent with their default configurations. Exploring the effectiveness of our method with alternative samplers in other models presents an exciting direction for future research. Additionally, thanks to this valuable suggestion, we conduct experiments on COCO with our model under different classifier-free guidance scales.  Note that cfg=7.0 is the default setting in the official implementation of SD3.
>
>
> **CLIP Score↑**
> | cfg  | SD3 28-steps | SD3 9-steps | RAPID3 |
> |------|--------------|-------------|--------|
> | 3.5  | 31.92        | 32.06       | 32.07  |
> | 5.0  | 32.05        | 32.14       | 32.08  |
> | 7.0(default)  | 32.06        | 31.88       | 32.09  |
> | 9.0  | 32.03        | 31.40       | 32.07  |
>
>
> **Aesthetic ↑**
> | cfg  | SD3 28-steps | SD3 9-steps | RAPID3 |
> |------|--------------|-------------|--------|
> | 3.5  | 5.18         | 5.08        | 5.16   |
> | 5.0  | 5.26         | 5.17        | 5.24   |
> | 7.0(default)  | 5.31         | 5.21        | 5.26   |
> | 9.0  | 5.35         | 5.18        | 5.37   |
>
> **Conclusion:**
> The results demonstrate the competitive performance of RAPID3 across various classifier-free guidance scales, highlighting its generality to different guidance strengths.

---

### Author Response · Authors · 2025-11-26
**To ACs and Reviewers**

Dear ACs and Reviewers,

We sincerely appreciate the time and effort you have dedicated to reviewing our submission. We are encouraged by the positive feedback: Reviewer Wtnj acknowledges that the proposed tri-level design is "**well motivated and complementary**", Reviewer 6idy finds our paper "**reasonable**" and the design "**clever**", and Reviewer ZdkW describes our method as "**conceptually elegant and practically impactful**".


We have carefully considered all constructive suggestions and have made every effort to address the concerns raised. We hope our revisions further enhance the quality of this submission.

Thank you once again for your valuable feedback.



Best regards

Authors of Submission 5605

---

### Meta-Review · Area_Chair_vBjs · 2026-01-07

**Summary:**

The submission proposes RAPID$^3$, a framework using Reinforcement Learning to train lightweight policy heads for accelerating frozen Diffusion Transformers via step-skipping, cache-reuse, and sparse-attention. The reviewers' initial concerns primarily focused on the comprehensiveness of the evaluation: specifically, the lack of comparisons to recent learned caching baselines (e.g., HarmoniCa), the reliance on limited metrics (CLIP/Aesthetic scores), and questions regarding the fairness of baseline hyperparameter tuning and inference overhead. The decision to accept is informed by the method's practical efficiency and the authors' robust rebuttal which filled these empirical gaps.

**Reviewer Concerns:**

Addressed:
- Missing Baselines (Reviewer ZdkW): The authors added a comparison with HarmoniCa, showing that RAPID$^3$'s tri-level strategy yields significantly better speed-quality trade-offs.
- Evaluation Metrics (Reviewer ZdkW): The authors integrated DPG-Bench results into the evaluation, providing a more rigorous assessment of fidelity and alignment beyond simple scores.
- Baseline Fairness & Overhead (Reviewer Wtnj): The rebuttal clarified that baselines were tuned for fair "equivalent step" comparisons and provided concrete measurements showing policy inference overhead is negligible (~2.2ms, <1% latency).
- Policy Interpretability (Reviewer 6idy, ZdkW): Additional visualizations (Appendix J) regarding the distribution of policy actions (e.g., distinguishing when cache vs. sparse attention is activated) successfully addressed the request for deeper insights into the learned synergy.
- Experimental Settings (Reviewer 6idy): The justification for using the 28-step default setting for SD3/FLUX was provided and aligns with standard community practices.

Outstanding:
- Extension to Video/Editing (Reviewer 6idy): The reviewer suggested extending validation to video diffusion or editing tasks. The authors acknowledged this as future work. This remains an open opportunity but is not considered a fatal flaw for the current scope.

**Reviewer Scores:**

- Reviewer Wtnj (Score: 6): Likely to remain 6. The reviewer was already positive ("marginally above acceptance"), and the detailed clarification regarding overhead, reproducibility, and sensitivity analysis addressed their specific technical queries.
- Reviewer 6idy (Score: 6): Likely to remain 6. The concerns were minor (justification of step count, surface-level ablation), and the rebuttal provided satisfactory explanations and deeper analysis.
- Reviewer ZdkW (Score: 4): Likely to increase to 6 . This reviewer's negative score was explicitly tied to the omission of relevant learned baselines and insufficient metrics. Since the authors explicitly ran the requested DPG-Bench experiments and compared against HarmoniCa with superior results, the basis for rejection has been resolved.

---

### Decision · Program_Chairs · 2026-01-26

Accept (Poster)